# Nano-topology optimization for materials design with atom-by-atom control

Chun-Teh Chen [1], Daryl C. Chrzan[1] & Grace X. Gu [2✉]

Atoms are the building blocks of matter that make up the world. To create new materials to meet some of civilization's greatest needs, it is crucial to develop a technology to design materials on the atomic and molecular scales. However, there is currently no computational approach capable of designing materials atom-by-atom. In this study, we consider the possibility of direct manipulation of individual atoms to design materials at the nanoscale using a proposed method coined "Nano-Topology Optimization". Here, we apply the proposed method to design nanostructured materials to maximize elastic properties. Results show that the performance of our optimized designs not only surpasses that of the gyroid and other triply periodic minimal surface structures, but also exceeds the theoretical maximum (Hashin–Shtrikman upper bound). The significance of the proposed method lies in a platform that allows computers to design novel materials atom-by-atom without the need of a predetermined design.

[1] Department of Materials Science and Engineering, University of California, Berkeley 94720 CA, USA. [2] Department of Mechanical Engineering, University of California, Berkeley 94720 CA, USA. ✉email: ggu@berkeley.edu

In 1959, Richard P. Feynman gave his famous talk entitled "There is Plenty of Room at the Bottom"[1], which has inspired the field of nanotechnology and nanoscience. Nanotechnology involves the manipulation of materials at the nanoscale and has had significant impact on multiple research directions such as drug delivery and tissue engineering in medicine[2] and solar cells in renewable energy applications[3]. Atoms are the building blocks of matter that make up the world. To create new materials to meet some of civilization's greatest needs, it is crucial to develop a technology to design materials on the atomic and molecular scales. However, decades after Feynman envisioned the development of nanomachines, machines constructed of single atoms and molecules, we are still not capable of realizing his original idea to create materials or machines with atom-by-atom control—arranging atoms one-by-one the way we want them. As of today, not only is there a lack of mature manufacturing technologies that can directly manipulate individual atoms to create materials but also there is no computational approach capable of designing materials atom-by-atom. The ability to design materials with atomic-level precision will unleash the full potential of matter and create immense opportunities across a wide range of scientific and engineering fields.

Recent advances in additive manufacturing have opened the gate to complex and multiscale architected materials[4–7]. For instance, the photopolymerization-based two-photon lithography (TPL) technique can print arbitrarily complex three dimensional (3D) structures with submicrometer resolution[8]. To fully leverage these new manufacturing technologies, it is essential to develop computational approaches capable of optimizing materials structures at different length scales to achieve desired properties. Conventional materials discovery and design processes largely rely on empirical, trial-and-error observations, which are expensive and time-consuming. While the rational design of materials is challenging, researchers have been turning to nature for inspiration. This design approach is referred to as biomimicry and has led to many innovations in materials design such as nacre-inspired nanocomposites with high strength and high toughness[9,10], conch shell-inspired composites with high impact resistance[11], and bone-inspired materials with high fracture toughness[12]. In addition to nature, researchers have been looking for inspiration from mathematics to design novel materials. For instance, triply periodic minimal surfaces (TPMS) have drawn tremendous attention in the materials science community. There is a widely held view that cellular materials created by TPMS may have superior properties due to their unique geometric features. Cellular materials are made up of a representative unit cell that is repeated throughout. Light-weight cellular materials have various properties beyond solid materials. From a mathematical point of view, TPMS structures are interesting as their surfaces have zero mean curvature and are characterized by local area-minimizing. In the TPMS family, the gyroid is the most widely studied structure and has been found in nature such as the wing scales of various butterflies. In recent years, extensive investigations on gyroid cellular materials have been reported to explain the physics underlying their mechanical[13–16], thermal[17], optical[18], and electromagnetic[19] properties.

Although nature or mathematics is an important source of inspiration, they should not be taken as the predominant guide to design materials as there is no one structure that works best for every purpose. Recently, machine learning (ML), a branch of artificial intelligence (AI), has been perceived as a promising tool to design novel materials[20–23]. Nevertheless, using ML models for the inverse design of materials with thousands of design variables is still an active field of reseach[24]. In major engineering industries, a more mature design approach referred to as topology optimization (TO) has been extensively implemented. TO provides

unrestricted design freedom and has been successfully applied to problems with more than a billion design variables[25]. The objective of TO is to search for optimal shapes and material distributions to maximize the performance of materials or structures such as aircraft and automotive components, buildings and bridges, and cellular materials[25–29]. Despite a large number of interesting shapes and designs that were proposed using TO, the design domains were always discretized using a finite element mesh. For this reason, conventional TO approaches using the finite element method (FEM) have been so far limited to the design of structures at the continuum scale and cannot be applied to design materials at the atomistic level.

Here, we aim to bring the world closer to realizing Feynman's vision with a de novo TO approach capable of designing materials at the nanoscale with atom-by-atom control. To distinguish our TO approach using atomistic modeling from the conventional TO approaches using FEM, we name the proposed method "Nano-Topology Optimization (Nano-TO)". In this study, we apply Nano-TO to design nanostructured materials to maximize elastic properties. Results show that the performance of our optimized designs not only surpasses that of the gyroid and other TPMS structures but also exceeds the theoretical maximum that is defined by the Hashin–Shtrikman (HS) upper bound[30]. We demonstrate that by optimizing the surface effect at the nanoscale using Nano-TO, the theoretical maximum of the bulk modulus can be exceeded. The significance of the proposed method lies in a platform that allows computers to design novel materials atom-by-atom without the need of a predetermined design. We envision that a broad array of novel nanomaterials and nanomachines with unprecedented performance can be designed using Nano-TO.

## Results

**Nano-topology optimization.** In this study, we consider the possibility of direct manipulation of individual atoms to design materials at the nanoscale. The flowchart of Nano-TO is shown in Fig. 1. The objective of Nano-TO is to search for the best possible atom distributions to maximize (or minimize) a desired property of nanostructured materials. Two types of atoms are considered: real and virtual. The atom type is allowed to switch during the optimization process. A sensitivity analysis is performed to calculate the sensitivity value of each atom in a design domain (see "Methods" section). Conceptually, the sensitivity analysis evaluates the contribution of each atom to the objective function (desired property) and this information is used to redistribute the atoms in the design domain. Afterwards, a sensitivity filtering technique is applied to modify the sensitivity value of each atom based on a weighted average of the sensitivity values of other atoms in a fixed neighborhood. The neighborhood region is defined by the filter radius. The purpose of applying the sensitivity filtering technique is to obtain the sensitivity values of virtual atoms (see "Methods" section). The real atoms with the lowest sensitivity values are considered the most inefficient and will be removed (converted to virtual atoms) in the next iteration. On the other hand, the virtual atoms with the highest sensitivity values will be added back (converted to real atoms) to the design domain. The number of real atoms to be converted to virtual atoms and that of virtual atoms to be converted to real atoms are controlled by the rejection and admission rates, respectively. As the initial structure consists of mostly real atoms, to reach the target volume fraction (relative density), the rejection rate has to be larger than admission rate in the first stage of the optimization process. The net rejection rate can be defined as the rejection rate minus admission rate. The smaller the net rejection rate, the more iterations are required to reach the target volume fraction.

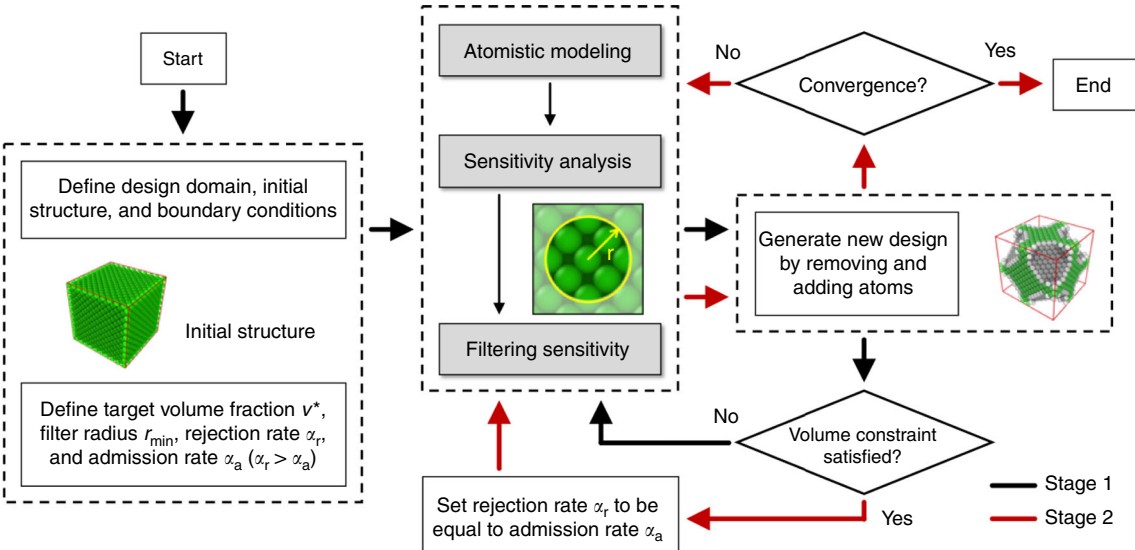

**Fig. 1 Flowchart of Nano-TO.** The flowchart shows the approach of Nano-TO to design materials at the nanoscale with atom-by-atom control. The black and red arrows represent the two optimization stages (stage 1 and 2), respectively.

However, if the net rejection rate is too large, it can cause the optimization to converge to a low-quality design or make the optimization process unstable. After the target volume fraction is reached, the rejection and admission rates are set to be equal in the second stage of the optimization process, until the optimization is converged. Lastly, the atoms denoted by real will be kept and the atoms denoted by virtual will be removed from the design domain. Consequently, a nanostructured material with an optimized atom distribution can be generated.

**Objective of maximizing the bulk modulus.** Nano-TO is a general-purpose design approach and can be applied to design materials at the nanoscale with different objectives. Here we apply Nano-TO to design nanostructured materials with the objective of maximizing the bulk modulus. The bulk modulus represents the resistance of a material to being elastically deformed by a hydrostatic pressure. Aluminum is selected as the base material and the initial structure is created based on the face-centered cubic (FCC) structure. The interactions between atoms are described by the embedded atom method (EAM)[31]. The design domain is a cubic unit cell with a length of approximately 4 nm. Periodic boundary conditions are imposed along the $x$-direction, $y$-direction, and $z$-direction to create the supercell structure for evaluating the macroscopic elastic properties. The initial structure consists of 4000 atoms. If all atoms are set to be real atoms and periodic boundary conditions are imposed, each atom in the design domain is identical and will have the same sensitivity value. For this reason, an atom at a random location is set to be a virtual atom to introduce an imperfection in the initial structure. For the optimization parameters, the target volume fraction is set to be in a range of 50–80%. In the first stage, to ensure the stability of the optimization process, the rejection and admission rates are set to be 2 and 1, respectively. Consequently, two real atoms with the lowest sensitivity values are converted to virtual atoms while a virtual atom with the highest sensitivity value is converted to a real atom in each iteration, until the target volume fraction is reached. In the second stage, the rejection and admission rates are both set to be 1, until the optimization is converged. The filter radius is chosen to be slightly smaller than the lattice constant of the base material (i.e., aluminum), which is approximately 4 Å. Consequently, the 12 nearest neighbors of each atom are considered when the sensitivity filtering technique

is implemented. Note that using a larger filter radius will cause the optimized designs to lose topological details (undesirable in this case) and increase the computational cost as more neighboring atoms have to be considered in the optimization process.

Nano-TO is a gradient-based design approach and the optimized designs would vary with the selection of the initial structure. To ensure that the optimized designs are of high performance (strong local minima), we use a strategy of random initialization (see "Methods" section). Starting with different initial structures, a total of 16 designs with a volume fraction of 50% are generated by Nano-TO. Those designs have the bulk modulus in a range of 18.25–22.20 GPa, with an average of 20.95 GPa. The design with the highest bulk modulus is denoted by Nano-TO design and selected for further examination. The Nano-TO designs with varying volume fractions are shown in Fig. 2a, b. Periodic images of the Nano-TO design with a volume fraction of 50% are shown in Fig. 2c and Supplementary Fig. 1. The structural evolution during the optimization process is presented in Supplementary Movies 1 and 2. It can be seen in the figures that the Nano-TO designs are nearly cubic symmetric. Furthermore, the virtual atoms form truncated octahedron structures in the body-centered cubic (BCC) arrangement. The surfaces in the Nano-TO designs are mostly {111} and {100}. Interestingly, the thermodynamically most stable structure (Wulff polyhedron) for a simple metal with a FCC structure (e.g., aluminum) is also the truncated octahedron exposing the faces {111} and {100}[32]. At present, it is unclear whether the optimal structures for maximizing the bulk modulus would be related to the Wulff polyhedron of the base material. Future studies are required to provide physical insight into this finding. Another case study for maximizing the elastic constant of $C_{33}$ is reported in Supplementary Notes, Supplementary Figs. 2–4, Supplementary Movies 3 and 4.

**Comparison of Nano-TO designs with TPMS structures.** To evaluate the capability of the proposed method, the Nano-TO designs are compared with TPMS structures including the gyroid, Schwarz D (diamond), and Schwarz P (primitive). Porous materials can be considered as two-phase composites with the solid and void phases. It has been shown that the HS bounds give an accurate estimate for the effective moduli of two-phase composites[30]. The HS upper bound is applied to calculate the

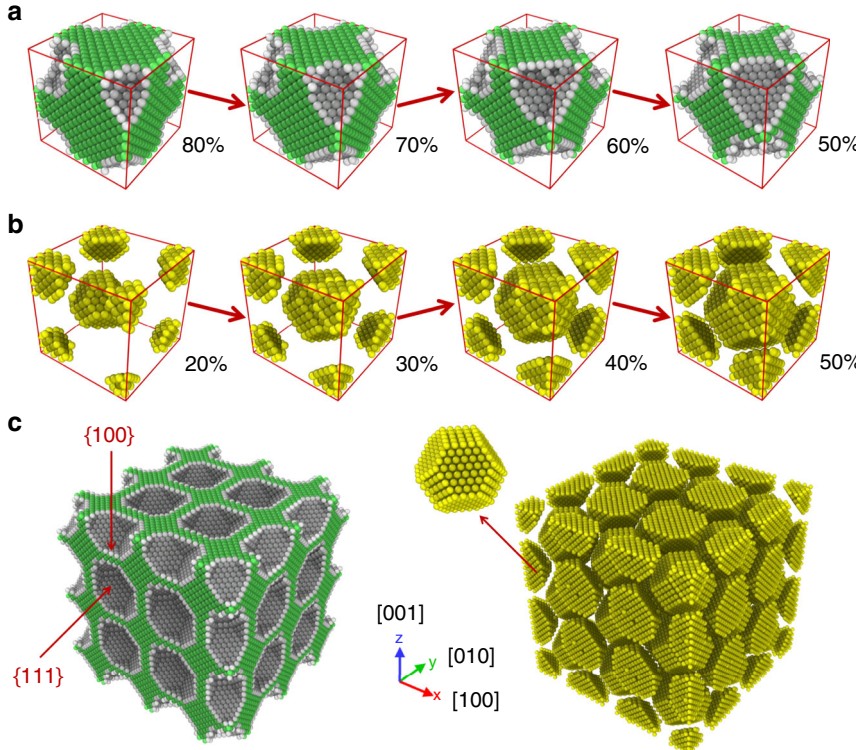

**Fig. 2 Nano-TO designs for maximizing bulk modulus. a** The Nano-TO designs for maximizing the bulk modulus with varying volume fractions from 50 to 80%. The bulk atoms are shown in green and the surface atoms are shown in gray. **b** The corresponding distributions of virtual atoms. **c** Periodic images (3 × 3 × 3) of the Nano-TO design with a volume fraction of 50% and the corresponding distribution of virtual atoms. The surfaces in the Nano-TO designs are mostly {111} and {100}. The virtual atoms form truncated octahedron structures in the BCC arrangement.

theoretical maxima of the bulk modulus for varying volume fractions (see "Methods" section). The bulk moduli of the Nano-TO designs and TPMS structures are shown in Fig. 3a. It can be seen in the figure that the bulk moduli of the Nano-TO designs surpass those of the TPMS structures with the same volume fraction. The result indicates that the TPMS structures are quite far from the optimal structures for maximizing the bulk modulus. Surprisingly, the bulk moduli of the Nano-TO designs exceed the HS upper bound. For instance, the bulk modulus of the Nano-TO design with a volume fraction of 50% is 22.20 GPa and the HS upper bound for the same volume fraction is only 19.63 GPa. We further confirm that the bulk moduli of the Nano-TO designs are indeed higher than the HS upper bound after considering the anisotropy effect (see "Methods" section and Supplementary Fig. 5).

**Exploring the surface effect**. The surface effect can no longer be neglected when the cell size of a material is reduced to a few nanometers. It is important to understand whether the Nano-TO designs are still superior to the TPMS structures at a larger scale. Here we take the Nano-TO design with a volume fraction of 50% as a template and parametrize the design based on vacancies with truncated octahedron structures in the BCC arrangement to create the same design with varying cell sizes. The Nano-TO design and TPMS structures with varying cell sizes from 4 to 64 nm are created (Supplementary Fig. 6) and their bulk moduli are shown in Fig. 3b. It can be seen in the figure that the bulk moduli of the Nano-TO design and TPMS structures vary with the cell size since the surface-to-volume ratio decreases with the cell size. This size dependence effect at the nanoscale provides an opportunity to design novel materials with superior properties. When

the cell size is 16 nm or larger, the surface effect in those structures is negligible. The bulk modulus of the Nano-TO design is always higher than those of the TPMS structures regardless of the cell size. Furthermore, the bulk modulus of the Nano-TO design exceeds the HS upper bound (19.63 GPa) when the cell size is small (e.g., 4 nm) and converges to the HS upper bound when the cell size is large enough. Hence the proposed method shows that the theoretical maximum bulk modulus is attainable in practice for this system and identifies a structure with that maximum bulk modulus. To shed light on the superiority of the Nano-TO design compared to the TPMS structures, the atomic strain ($\varepsilon_{zz}$) distributions of those structures subjected to a constant hydrostatic strain of $-10^{-2}$ are shown in Fig. 3c. The cell size in the comparison is 16 nm and the volume fraction is 50%. The average atomic strains for the Nano-TO design, gyroid, diamond, and primitive are $-0.0048$, $-0.0033$, $-0.0035$, and $-0.0029$, respectively. The average atomic strains of $\varepsilon_{xx}$ and $\varepsilon_{yy}$ are the same as that of $\varepsilon_{zz}$ due to the symmetry of those structures. The result shows that the Nano-TO design has a better load transfer mechanism as the average atomic strain (compressive) is larger than those of the TPMS structures. This is probably due to the structural simplicity of the Nano-TO design compared to the complex TPMS structures. Consequently, a higher bulk modulus is achieved as more elastic strain energy can be accommodated. Note that the TPMS structures in the comparison are based on solid-networks. Another comparison with the TPMS structures based on sheet-networks is reported in Supplementary Notes, Supplementary Figs. 7 and 8.

**Exceeding the theoretical maximum**. To understand how the bulk modulus of a material can exceed the theoretical maximum,

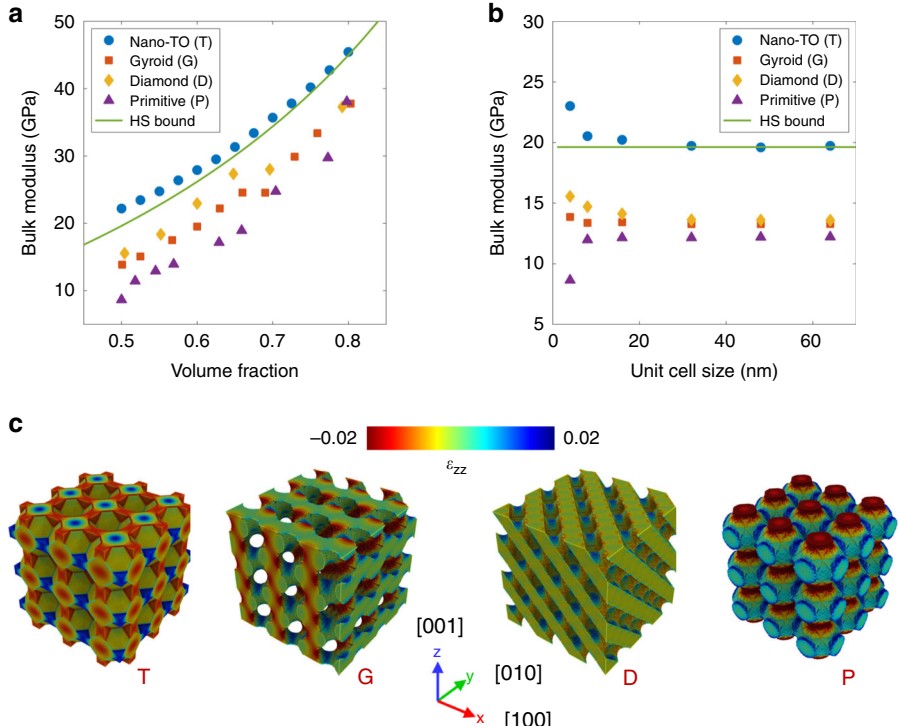

**Fig. 3 Performance of Nano-TO design and TPMS structures. a** The bulk moduli of the Nano-TO design (blue circles), gyroid structure (red squares), diamond structure (yellow diamonds), and primitive structure (purple triangles) with varying volume fractions in comparison with the HS upper bound (green line). The cell size in the comparison is 4 nm. **b** The bulk moduli of the Nano-TO design (blue circles), gyroid structure (red squares), diamond structure (yellow diamonds), and primitive structure (purple triangles) with varying cell sizes from 4 to 64 nm in comparison with the HS upper bound (green line). The volume fraction in the comparison is 50%. **c** Atomic strain ($\varepsilon_{zz}$) distributions of those structures subjected to a constant hydrostatic strain of $-10^{-2}$. The cell size of those structures is 16 nm and the volume fraction is 50%. The average atomic strains for the Nano-TO design (T), gyroid (G), diamond (D), and primitive (P) are $-0.0048$, $-0.0033$, $-0.0035$, and $-0.0029$, respectively.

slice views of the Nano-TO design with a volume fraction of 50% are shown in Fig. 4a to reveal the interconnectivity. It can be seen in the figure that the Nano-TO design consists of nanoplate structures with {111} surfaces. Depending on the cell size, the nanoplate structures consist of varying numbers of surface and bulk layers. For instance, in the unit cell with a length of 4 nm, the nanoplate structures are three atomic-layer thick, including two surface layers and one bulk layer. When the cell size is increased to 16 nm, the nanoplate structures become 12 atomic-layer thick, including 2 surface layers and 10 bulk layers. Elastic properties of the surface material are different from those of the bulk material[33]. To quantify the differences, an atomistic model of a nanoplate with a free surface {111} on each side is created and shown in Fig. 4b. Periodic boundary conditions are imposed along the in-plane directions (i.e., $[1\bar{1}0]$ and $[11\bar{2}]$). Vacuum regions are created for the top and bottom of the nanoplate to represent two flat free surfaces. In this specific model, the nanoplate consists of 12 layers including 2 surface layers and 10 bulk layers. Other models with varying numbers of layers from 3 to 48 are also created. We first investigate extreme cases with the minimum and maximum thicknesses. The nanoplate consists of three layers including two surface layers and one bulk layer are considered as the thinnest possible nanoplate and is denoted by surface model. On the other hand, the nanoplate with the infinite number of layers, denoted by bulk model, is created by removing the vacuum regions and applying a periodic boundary condition along the out-of-plane direction (i.e., [111]). We apply normal strains along the two in-plane directions in a range of $-10^{-2}$ to $10^{-2}$ on those two models and the corresponding strain–energy curves are shown in Fig. 4c.

The result shows that the material with free surfaces is elastically stiffer, indicating that the {111} surfaces are stiffer than the bulk material in the $[1\bar{1}0]$ and $[11\bar{2}]$ directions. Furthermore, the Young's moduli of nanoplates in the in-plane directions are the same since the strain–energy curves in the $[1\bar{1}0]$ and $[11\bar{2}]$ directions are identical. To quantify how much stiffer the material with free surfaces is compared to the bulk material, the modulus ratios in the two in-plane directions for the nanoplates with varying numbers of layers are shown in Fig. 4d. The modulus ratio is defined as the Young's modulus of a nanoplate divided by that of the bulk material. The result shows that the nanoplate consisting of three layers is the stiffest, which is more than 30% stiffer than the bulk material. Note that surfaces can be softer than the bulk material as the lower atomic coordination on surfaces tends to make them softer (see Supplementary Notes). However, in certain combinations of surface orientation and loading direction, the electron redistribution on surfaces may give rise to stronger bonding, and thus making them stiffer than the bulk material[34]. Whether a surface is stiffer or softer depends on the competition between the electron redistribution and the lower atomic coordination on the surface.

## Discussion

We demonstrate that Nano-TO can utilize the surface effect in the design of nanostructured materials. By optimizing the surface topology at the nanoscale, the HS upper bound for the bulk modulus can be exceeded. Note that the surface effect comes from the variation of bond strength with coordination, which can only be captured in atomistic simulations. Therefore, the conventional

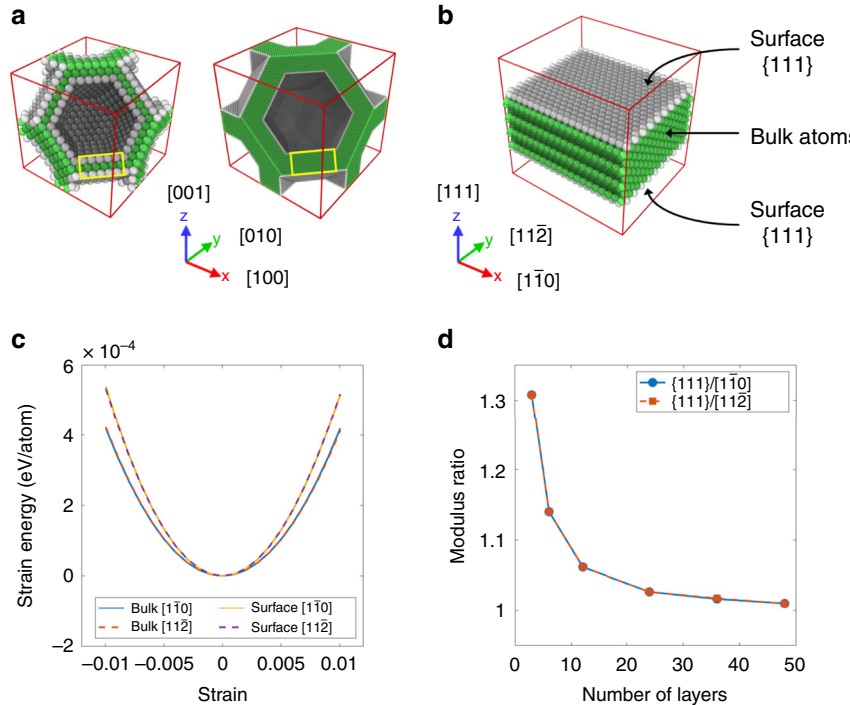

**Fig. 4 Surface effect in nanostructured materials. a** Slice views of the Nano-TO design with a volume fraction of 50% to reveal the interconnectivity. The bulk atoms are shown in green and the surface atoms are shown in gray. The length of the unit cell on the left is 4 nm and that on the right is 16 nm. The yellow boxes indicate a nanoplate structure with surfaces {111}. **b** The atomistic model of a nanoplate with surfaces {111}. **c** Strain–energy curves for the bulk model in the $[1\bar{1}0]$ (blue solid line) and $[11\bar{2}]$ (red dashed line) directions, and for the surface model in the $[1\bar{1}0]$ (yellow solid line) and $[11\bar{2}]$ (purple dashed line) directions. **d** The modulus ratios in the $[1\bar{1}0]$ (blue circles with solid line) and $[11\bar{2}]$ (red squares with dashed line) directions for the nanoplates with varying numbers of layers. The modulus ratio is defined as the modulus of a nanoplate divided by that of the bulk material.

TO approaches using FEM will not generate the Nano-TO designs shown in this study. We show the applications of Nano-TO on the design of nanostructured materials for maximizing a desired property, in which two objectives, the bulk modulus and elastic constant of $C_{33}$ (see Supplementary Notes), are considered, respectively. Various multiobjective optimization methods[35] can be implemented in Nano-TO to design nanostructured materials with multiple desired properties. The most common approach is the weighted sum method. To apply this method, the sensitivity values of each atom for different objectives are calculated individually and the weighted sensitivity value is calculated by choosing proper weights (user's preference) for different objectives. Consequently, optimized designs with the best tradeoff between competing objectives can be generated by Nano-TO. Although the examples presented here focus on elastic moduli, Nano-TO can be applied to design nanomaterials with other properties. It has been shown that the mechanical properties of metallic materials are highly related to their elastic moduli. For instance, an elastic anisotropy parameter can be used to identify alloys that display super elasticity, super strength, and high ductility, known as gum metals[36]. If the correlations between the elastic moduli of a material and its other mechanical properties (e.g., failure strain, strength, and toughness) can be discovered, the same design approach to tailor materials' elastic moduli can be applied to tailor materials' other mechanical properties.

The reliability of Nano-TO depends on the accuracy of the atomistic modeling implemented in the optimization process. In this study, we choose aluminum as the base material since there are interatomic potentials available in the literature to accurately reproduce basic equilibrium properties of aluminum including its elastic constants, vacancy formation and migration energies, and surface energies. With accurate interatomic potentials, Nano-TO

can be applied to design nanostructured materials using other base materials (e.g., copper, nickel, and gold). Most TO problems at the continuum-scale, except for some simple cases, are non-convex optimization problems, which contain many local minima[29]. We find that this is also the case in the Nano-TO examples presented in this study since the optimized designs are generally dependent on the initial structure. We show that Nano-TO is capable of identifying the optimal designs by using several different initial structures. However, this dependence on the initial structure could become a computational bottleneck when applying Nano-TO to large-scale materials design problems as the computational cost is increased. Therefore, further improvements of the Nano-TO approach are essential to make it less sensitive to the initial structure and prevent generating low-quality designs.

A typical material with a volume of a few cubic centimeters consists of around $10^{23}$ atoms. Even if the volume is reduced to a few cubic micrometers, the material still consists of around $10^{11}$ atoms. To achieve atomic-level precision, each atom is a design variable. Therefore, the computational cost to design materials on a scale of only a few micrometers is already beyond current computational capabilities. However, this computational bottleneck could be overcome in the future using AI and ML techniques. Future studies are required to develop suitable ML techniques to replace the computationally expensive atomistic modeling in the sensitivity analysis. We envision that this ML-based approach could potentially reduce the computational cost of Nano-TO by several orders-of-magnitude.

## Methods
**Cellular materials based on TPMS**. Three lattice types based on TPMS are investigated: the gyroid (G), diamond (D), and primitive (P). Their level surface

approximations are[37]:

$$U_G = \sin\left(2\pi\frac{x}{a}\right)\cos\left(2\pi\frac{y}{a}\right) + \sin\left(2\pi\frac{y}{a}\right)\cos\left(2\pi\frac{z}{a}\right) + \sin\left(2\pi\frac{z}{a}\right)\cos\left(2\pi\frac{x}{a}\right) - t, \tag{1}$$

$$U_D = \sin\left(2\pi\frac{x}{a}\right)\sin\left(2\pi\frac{y}{a}\right)\sin\left(2\pi\frac{z}{a}\right) + \sin\left(2\pi\frac{x}{a}\right)\cos\left(2\pi\frac{y}{a}\right)\cos\left(2\pi\frac{z}{a}\right) + \cos\left(2\pi\frac{x}{a}\right)\sin\left(2\pi\frac{y}{a}\right)\cos\left(2\pi\frac{z}{a}\right) + \cos\left(2\pi\frac{x}{a}\right)\cos\left(2\pi\frac{y}{a}\right)\sin\left(2\pi\frac{z}{a}\right) - t, \tag{2}$$

$$U_P = \cos\left(2\pi\frac{x}{a}\right) + \cos\left(2\pi\frac{y}{a}\right) + \cos\left(2\pi\frac{z}{a}\right) - t, \tag{3}$$

where $a$ is the length of a cubic unit cell and $t$ is the threshold of the surface. The surface is generated by finding $U = 0$. Unit cells with a finite volume are created by in-filling one side of the surface. When $t$ is 0, a porous structure with a relative density (volume fraction) of 0.5 will be created. Porous structures with varying volume fractions can be created by adjusting the value of $t$ (Supplementary Table 1). The maximum volume fraction is set to be 80% (i.e., 80% solid phase and 20% void phase). The minimum volume fraction is set to be 50% to avoid instability of porous structures due to a large portion of surface atoms in a small unit cell.

**Sensitivity analysis**. Pair potentials (e.g., Lennard–Jones) do not include the environmental dependence of bonding. Therefore, the strength of individual bonds in the bulk is the same as that on (or near) the surface, which is physically not true. This local environmental dependence is especially important for simulations of surfaces and can be considered in many-body potentials such as the EAM potentials[31]. The total potential energy of a material system consisting of $N$ atoms in the EAM potentials is given by:

$$E_{tot} = \sum_{i=1}^{N}\left\{ F_\alpha\left(\sum_{j\neq i}^{N}\rho_\beta(r_{ij})\right) + \frac{1}{2}\sum_{j\neq i}^{N}\phi_{\alpha\beta}(r_{ij})\right\}, \tag{4}$$

where $r_{ij}$ is the distance between atoms $i$ and $j$, $F_\alpha$ is the embedding energy to place atom $i$ of type $\alpha$ into the electron cloud, $\rho_\beta$ is the contribution to the electron charge density from atom $j$ of type $\beta$ at the location of atom $i$, $\phi_{\alpha\beta}$ is the pair-wise potential energy between atom $i$ of type $\alpha$ and atom $j$ of type $\beta$. Since the embedding energy term considers the local background electron density of atoms, the EAM potentials can describe the variation of bond strength with coordination. Therefore, the EAM potentials are applicable for modeling material systems with surfaces or other crystalline defects as those investigated in this study. To adopt an EAM potential in Nano-TO, the potential is modified in the way that the design variables are integrated into the total potential energy of a material system. The modified EAM potential is:

$$E_{tot} = \sum_{i=1}^{N}\left\{ x_i F_\alpha\left(\sum_{j\neq i}^{N}x_j\rho_\beta(r_{ij})\right) + \frac{1}{2}x_i\sum_{j\neq i}^{N}x_j\phi_{\alpha\beta}(r_{ij})\right\}, \tag{5}$$

where $x$ is a design variable, in which 1 indicates that the corresponding atom is kept and 0 indicates that the atom is removed. Here, instead of removing atoms from a material system, atoms are converted to virtual atoms. Therefore, those atoms with the design variable of 1 are referred to as real atoms and the others with the design variable of 0 are referred to as virtual atoms. When a material system is subjected to a constant external strain (e.g., $\varepsilon_{zz}$), the elastic strain energy of the system can be calculated as the energy difference between the total potential energy in the deformed state and that in the equilibrium state. The calculation is:

$$E_{ela} = \sum_{i=1}^{N}\left\{ x_i F_\alpha\left(\sum_{j\neq i}^{N}x_j\rho_\beta(r_{ij,def})\right) + \frac{1}{2}x_i\sum_{j\neq i}^{N}x_j\phi_{\alpha\beta}(r_{ij,def})\right\} - \sum_{i=1}^{N}\left\{ x_i F_\alpha\left(\sum_{j\neq i}^{N}x_j\rho_\beta(r_{ij,equ})\right) + \frac{1}{2}x_i\sum_{j\neq i}^{N}x_j\phi_{\alpha\beta}(r_{ij,equ})\right\}, \tag{6}$$

where $r_{ij,def}$ is the distance between atoms $i$ and $j$ in the deformed state and $r_{ij,equ}$ is the distance between atoms $i$ and $j$ in the equilibrium state. Compared to the pair-wise potential energy $\phi_{\alpha\beta}$, the embedding energy $F_\alpha$ is less sensitive to a change of $r_{ij}$ (see Supplementary Notes, Supplementary Figs. 9 and 10). Thus, to simplify the calculation, the elastic strain energy of the material system is approximated as:

$$E_{ela} \sim \sum_{i=1}^{N}\left\{ \frac{1}{2}x_i\sum_{j\neq i}^{N}x_j\phi_{\alpha\beta}(r_{ij,def})\right\} - \sum_{i=1}^{N}\left\{ \frac{1}{2}x_i\sum_{j\neq i}^{N}x_j\phi_{\alpha\beta}(r_{ij,equ})\right\}, \tag{7}$$

Assuming that the objective is to maximize the elastic constant of $C_{33}$. In this case, the elastic constant of $C_{33}$ is associated with the external strain of $\varepsilon_{zz}$. Thus, maximizing the elastic constant of $C_{33}$ is equivalent to maximizing the elastic strain energy of the material system subjected to a constant external strain of $\varepsilon_{zz}$. This

approach can also be applied to optimize other elastic properties. For instance, maximizing the bulk modulus is equivalent to maximizing the elastic strain energy of the material system subjected to a constant hydrostatic strain. The optimization problem can be written as:

$$\max_{\mathbf{x}}\left(\sum_{i=1}^{N}\left\{\frac{1}{2}x_i\sum_{j\neq i}^{N}x_j\phi_{\alpha\beta}(r_{ij,def})\right\} - \sum_{i=1}^{N}\left\{\frac{1}{2}x_i\sum_{j\neq i}^{N}x_j\phi_{\alpha\beta}(r_{ij,equ})\right\}\right),$$
$$\text{subject to}: \sum_{i=1}^{N}\frac{x_i}{N} = v^* \tag{8}$$

where $v^*$ is the prescribed volume fraction. Nano-TO uses a sensitivity analysis to determine the atom rejection and admission. The gradient of the objective function (i.e., elastic strain energy) with respect to the design variable $x_k$ can be approximated as:

$$\frac{\partial E_{ela}}{\partial x_k} \sim \frac{\partial}{\partial x_k}\sum_{i=1}^{N}\left\{\frac{1}{2}x_i\sum_{j\neq i}^{N}x_j\phi_{\alpha\beta}(r_{ij,def})\right\} - \frac{\partial}{\partial x_k}\sum_{i=1}^{N}\left\{\frac{1}{2}x_i\sum_{j\neq i}^{N}x_j\phi_{\alpha\beta}(r_{ij,equ})\right\}$$
$$=\left\{\frac{1}{2}\sum_{j\neq k}^{N}x_j\phi_{\alpha\beta}(r_{kj,def}) + \frac{1}{2}\sum_{i\neq k}^{N}x_i\phi_{\alpha\beta}(r_{ik,def})\right\}$$
$$-\left\{\frac{1}{2}\sum_{j\neq k}^{N}x_j\phi_{\alpha\beta}(r_{kj,equ}) + \frac{1}{2}\sum_{i\neq k}^{N}x_i\phi_{\alpha\beta}(r_{ik,equ})\right\}$$
$$=\sum_{j\neq k}^{N}x_j\phi_{\alpha\beta}(r_{kj,def}) - \sum_{i\neq k}^{N}x_i\phi_{\alpha\beta}(r_{ik,equ}) \sim 2\frac{E_{k,ela}}{x_k}, \tag{9}$$

where $E_{k,ela}$ is the elastic strain energy of atom $k$. The design variable $x_k$ can either be 0 or 1. If atom $k$ is a real atom ($x_k = 1$), its sensitivity value (gradient) can be approximated as two times of its elastic strain energy. On the other hand, if atom $k$ is a virtual atom ($x_k = 0$), its sensitivity value is undefined.

**Sensitivity filtering**. TO problems at the continuum-scale using FEM have shown that modifying the sensitivity value of each element (finite element) based on a weighted average of the element sensitivity values in a fixed neighborhood is a highly efficient way to ensure mesh-independency[38]. The filtered sensitivity value of atom $k$ is calculated as:

$$\widehat{\frac{\partial E_{ela}}{\partial x_k}} = \frac{1}{\sum_{i=1}^{n}\hat{H}_i}\sum_{i=1}^{n}\hat{H}_i\frac{\partial E_{ela}}{\partial x_k},$$
$$\hat{H}_i = r_{min} - \text{dist}(k,i), \{i \in n | \text{dist}(k,i) \leq r_{min}\} \tag{10}$$

where $n$ is the total number of atoms (including virtual atoms) in a fixed neighborhood of atom $k$, $\hat{H}_i$ is the weighting factor for atom $i$, $\text{dist}(k,i)$ is the distance between atoms $k$ and $i$, and $r_{min}$ is the filter radius. Note that the sensitivity value of a virtual atom is undefined. Thus, it is set to be zero in the sensitivity analysis. However, the filtered sensitivity value of a virtual atom could be nonzero when there are real atoms in its fixed neighborhood. This nonzero sensitivity information for virtual atoms is essential as it allows Nano-TO to determine which virtual atoms should be converted to real atoms in the optimization process.

**Atomistic modeling**. Full-atomistic simulations are implemented using Large-scale Atomic/Molecular Massively Parallel Simulator (LAMMPS)[39]. A modified EAM potential for aluminum is adopted for the simulations. The original version of the EAM potential was developed by Mishin et al.[40]. The potential functions of the embedding energy $F_\alpha$, atomic electron density $\rho_\beta$, and pair-wise potential energy $\phi_{\alpha\beta}$ in the EAM potential are fitted to both experimental data and ab initio calculations to accurately reproduce basic equilibrium properties of aluminum including its elastic constants, vacancy formation and migration energies, and surface energies. To give Nano-TO the ability to remove an atom from a material system and add it back when it is needed, a new atom type denoted by virtual is introduced. When a real atom is converted to a virtual atom, the interactions between the virtual atom and the other atoms in the material system are completely turned off. Physically, the atom is removed from the material system as it does not interact with the other atoms at all. However, the position information of the virtual atom is reserved in order to convert the virtual atom back to a real atom when it is needed. To ensure that the modified EAM potential and elastic properties calculations are accurate, we compare the lattice properties of aluminum predicted in this study with those reported in the literature[40]. The comparison is shown in Supplementary Table 2 and the results are identical. The approach to calculate the elastic properties is described in the Supplementary Methods. The Open Visualization Tool (OVITO)[41] is implemented for the visualization and atomic strain analysis.

**Hashin–Shtrikman (HS) bounds**. The HS bounds provide theoretical upper and lower bounds for the effective elastic moduli of multiphase materials of arbitrary phase geometry[30]. The HS bounds for the bulk modulus of an, at least cubic

symmetric, two-phase composite are given by:

$$K_{HS}^{+} = K_2 + \frac{v_1}{(K_1 - K_2)^{-1} + 3v_2(3K_2 + 4G_2)^{-1}}, \quad (11)$$

$$K_{HS}^{-} = K_1 + \frac{v_2}{(K_2 - K_1)^{-1} + 3v_1(3K_1 + 4G_1)^{-1}}, \quad (12)$$

where $K_{HS}^{+}$ and $K_{HS}^{-}$ represent the upper and lower bounds, respectively. $K$ is the bulk modulus, $G$ is the shear modulus, and $v$ is the volume fraction. The subscript of $K$, $G$, and $v$ denotes the phase of materials. The phase 2 material is assumed to be stiffer than the phase 1 material ($K_2 > K_1$; $G_2 > G_1$). For porous materials, the solid phase can be represented as the phase 2 material and the void phase as the phase 1 material ($K_1 = 0$; $G_1 = 0$). The HS upper bound for the bulk modulus of a porous material is given by:

$$K_{HS}^{+} = K_2 + \frac{v_1}{(-K_2)^{-1} + 3v_2(3K_2 + 4G_2)^{-1}}. \quad (13)$$

The HS lower bound for the bulk modulus of a porous material is zero. The Voigt–Reuss–Hill (VRH) average[42] is applied to compute the shear modulus of aluminum (nearly isotropic). This approach calculates the mean value of the upper and lower bounds of the effective shear modulus obtained through the Voigt and Reuss assumptions:

$$G_V = \frac{(C_{11} - C_{12}) + 3C_{44}}{5}, \quad (14)$$

$$G_R = \frac{5C_{44}(C_{11} - C_{12})}{4C_{44} + 3(C_{11} - C_{12})}, \quad (15)$$

$$G_{avg} = \frac{G_V + G_R}{2}, \quad (16)$$

where $G_V$ and $G_R$ represent the upper and lower bounds, respectively. $G_{avg}$ is the average shear modulus. The elastic constants used in the calculations are reported in Supplementary Table 2. The average shear modulus (29.28 GPa) is used to calculate the HS upper bound for the bulk modulus.

To eliminate the anisotropy effect of aluminum and ensure that the HS upper bound for the bulk modulus can certainly be surpassed by utilizing the surface effect at the nanoscale, the range of the shear modulus of aluminum is investigated. For an anisotropic material, the shear modulus varies with the shear plane as well as the shear direction. The transformation of the shear modulus from a reference set of axes (1, 2, and 3) to a new set of axes (1′, 2′, and 3′) can be written as[43]:

$$G_4' = \frac{1}{s_{44}'}, \quad (17)$$

$$s_{44}' = s_{44} + (4s_{11} - 4s_{12} - 2s_{44})(a_{31}^2 a_{21}^2 + a_{32}^2 a_{22}^2 + a_{33}^2 a_{23}^2), \quad (18)$$

where $G_4'$ is the shear modulus for the new set of axes, $s_{ij}$ is the elastic compliance constant using the Voigt notation, and $a_{im}$ is the direction cosine indicating the angle between the $i$ axis of the new axis system and the $m$ axis of the reference axis system. Here, without loss of generality, the new axis 3′ is set to be the plane of shear and the new axis 2′ is set to be the direction of shear. After exploring all possible combinations of the shear plane and direction, the maximum shear modulus ($G_{max}$) and minimum shear modulus ($G_{min}$) of aluminum are identified. $G_{max}$ is calculated as 31.60 GPa, which is associated with a shear in the ⟨100⟩ direction and is independent of the shear plane. $G_{min}$ is calculated as 26.12 GPa, which is associated with a shear in the ⟨110⟩ direction on a {110} plane. $G_{max}$ and $G_{min}$ represent the extreme values of the shear modulus of aluminum. Thus, the HS upper bounds calculated with these two extreme values represent the limits when taking the anisotropy effect into consideration.

## Data availability
All data needed to evaluate the conclusions in this study are present in the paper and Supplementary Information. Additional data related to this study are available from the corresponding author upon reasonable request.

## Code availability
All necessary information to generate the code used to evaluate the conclusions in this study are present in the paper and Supplementary Information. The code used for the atomistic simulations is the open-source code, LAMMPS. The interatomic potential can be found at http://www.ctcms.nist.gov/potentials. The script used to calculate elastic constants can be found at https://github.com/lammps/lammps/tree/master/examples/ELASTIC.

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

## Acknowledgements

We acknowledge support from Extreme Science and Engineering Discovery Environment (XSEDE) Bridges system by the National Science Foundation (grant number ACI-1548562), NVIDIA GPU Seed Grant, Haythornthwaite Foundation Research Initiation Grant, and the Savio computational cluster resource provided by the Berkeley Research Computing program.

## Author contributions

C.-T.C., D.C.C., and G.X.G conceived the idea. C.-T.C. designed the theory and modeling approach, implemented the simulations, and analyzed the data. C.-T.C. wrote the manuscript with valuable input from D.C.C. and G.X.G.

## Competing interests

The authors declare no competing interests.
