## [Peer Review File · Nature Communications]

REVIEWER COMMENTS

Reviewer #1 (Remarks to the Author):

In this paper, the authors propose a design method on Topology Optimization that can search for the best possible atom distributions for desired properties of nanostructured materials. Through designing Al structures atom-by-atom without the need of a predetermined design, the elastic properties such as the bulk modulus can exceed the theoretical maximum. The resulting performance of the optimized design also surpasses that of other triply periodic minimal surface structures. These results are of interest to the readers in the related field of advanced design for nanomaterials and nanomachines with desired properties. I have several suggestions and I hope that the authors can address these points before the publication of this manuscript.

1. In Figure 1 and supporting information, the concept of rejection and admission rate and how the value is selected in the optimization process have not been discussed. The definition for the filter radius in Eq. 10 should also be provided. How do these parameters affect the optimization?
2. In this work, aluminum is selected as the base material. How about the application of the method in other material systems?
3. In the supporting information, the nanoplate with a free surface $\{1\ 1\ 0\}$ is selected for comparison. How about other faces?
4. The authors present the case studies of using Nano-TO in maximizing the bulk modulus and elastic constant. If the desired performance of nanostructured materials requires two or more improved mechanical properties, how to make use of this method and make balance in the design?
5. As the authors mentioned that “the generality of the approach that it can be used to design nanomaterials and nanomachines with optimal performance”, it would be helpful to compare this method with prior approaches such as mentioned ML models, and describe how their method bases on the TO can be transferred to other nanomachines.

Reviewer #2 (Remarks to the Author):

The work presents a method to optimize atomic structures. Though the algorithm is simple and straightforward, the concept is innovative. The following questions should be well answered before acceptance:

1)The process of optimization is shown in the flow chart in Fig. 1 but has not well been explained.

2)A robust optimization algorithm should not rely on the initial structures. But the authors claim that 16 structures were produced in 16 attempts with random initialization. Then how the optimization guarantee the minimum of objective functional, no matter locally or globally.

3)Is the surface effect considered in atomistic modeling? If yes, how to take it into consideration?

4)How to impose periodic boundary condition?

5)How to determine the embedding energy and potential energy in Eq. (4).

6)The statement “an energy minimization using the conjugate gradient (CG) algorithm is performed to equilibrate the material system” is very unclear. Please give more details to show how to calculate the elastic properties? In homogenization theory when calculating the effective properties like C_{11} , only one test strain is imposed. But the work needs a positive strain and a negative test strain, why? Will the magnitude of test strain affect the value of C_{11} ?

Reviewer #3 (Remarks to the Author):

The paper presents a so-called nano-topology optimization method for atom-by-atom controlled material design. While the paper is well written and well described the method, the reviewer didn't find it interesting.

In particular, I couldn't find any novelty and broader impacts of the paper.

It seems the authors are trying to re-invent the wheel that is already out there—by using the same old-fashioned optimization approach, just at a different scale.

More importantly, people in the design and optimization community have already tackled these kinds of problems in many different ways and at different length scales. A small survey on the designing (and even manufacturing) of architected and cellular material with desired properties (linear/nonlinear) would reveal those well-established approaches.

Therefore, I don't see any novelty and broader impact by re-trying a well-established old-fashioned topology optimization approach at another length scale. To me, this is just a matter of scaling, and you could get the same results (even better) at the continuum level. You could also develop any of these objectives (even complex cases) at the continuum level and obtain any desired properties, as people have done this before.

In conclusion, the reviewer does not recommend the manuscript for publication in Nat. Comm.

Response to Reviewers' Comments

Paper title: Nano-topology optimization for materials design with atom-by-atom control

Authors: Chun-Teh Chen, Daryl C. Chrzan, Grace X. Gu

We thank the reviewers for their insightful comments and critique of the presented work. Please find below a point-by-point response to every comment and indications to what and where changes have been made. These changes are indicated in blue text in the revised paper.

REVIEWER REPORTS:

Reviewer 1

In this paper, the authors propose a design method on Topology Optimization that can search for the best possible atom distributions for desired properties of nanostructured materials. Through designing AI structures atom-by-atom without the need of a predetermined design, the elastic properties such as the bulk modulus can exceed the theoretical maximum. The resulting performance of the optimized design also surpasses that of other triply periodic minimal surface structures. These results are of interest to the readers in the related field of advanced design for nanomaterials and nanomachines with desired properties. I have several suggestions and I hope that the authors can address these points before the publication of this manuscript.

1) In Figure 1 and supporting information, the concept of rejection and admission rate and how the value is selected in the optimization process have not been discussed. The definition for the filter radius in Eq. 10 should also be provided. How do these parameters affect the optimization?

We thank the reviewer for this suggestion. We have added conceptual descriptions of the rejection and admission rates as well as the filter radius to the Results section where Figure 1 is mentioned:

“Conceptually, the sensitivity analysis evaluates the contribution of each atom to the objective function (desired property) and this information is used to redistribute the atoms in the design domain. Afterwards, a sensitivity filtering technique is applied to modify the sensitivity value of each atom based on a weighted average of the sensitivity values of other atoms in a fixed neighborhood. The neighborhood region is defined by the filter radius. The purpose of applying the sensitivity filtering technique is to obtain the sensitivity values of “virtual” atoms (see Methods).”

“The number of “real” atoms to be converted to “virtual” atoms and that of “virtual” atoms to be converted to “real” atoms are controlled by the rejection and admission rates, respectively. As the initial structure consists of mostly “real” atoms, to reach the target volume fraction (relative density), the rejection rate has to be larger than admission rate in the first stage of the optimization process. The net rejection rate can be defined as the rejection rate minus admission rate. The smaller the net rejection rate, the more iterations are required to reach the target volume fraction. However, if the net rejection rate is too large, it can cause the optimization to converge to a low-quality design or make the optimization process unstable. After the target volume fraction is reached, the rejection and admission rates are set to be equal in the second stage of the optimization process, until the optimization is converged. Lastly, the atoms denoted by “real” will be kept and the atoms denoted by “virtual” will be removed from the design domain. Consequently, a nanostructured material with an optimized atom distribution can be generated.”

Additionally, we have added discussions of how the optimization parameters are selected and their effects in the optimization process to the Results section where the optimization setup is described:

“In the first stage, to ensure the stability of the optimization process, the rejection and admission rates are set to be 2 and 1, respectively. Consequently, two “real” atoms with the lowest sensitivity values are converted to “virtual” atoms while a “virtual” atom with the highest sensitivity value is converted to a “real” atom in each iteration, until the target volume fraction is reached. In the second stage, the rejection and admission rates are both set to be 1, until the optimization is converged. The filter radius is chosen to be slightly smaller than the lattice constant of the base material (i.e., aluminum), which is approximately 4 Å. Consequently, the 12 nearest neighbors of each atom are considered when the sensitivity filtering is implemented. Note that using a larger filter radius will cause the optimized designs to lose topological details (undesirable in this case) and increase the computational cost as more neighboring atoms have to be considered in the optimization process.”

2) In this work, aluminum is selected as the base material. How about the application of the method in other material systems?

We appreciate the reviewer for pointing this out. We have added discussions of the application of Nano-TO in other material systems to the Discussions section:

“The reliability of Nano-TO depends on the accuracy of the atomistic modeling implemented in the optimization process. In this study, we choose aluminum as the base material since there are interatomic potentials available in the literature to accurately reproduce basic equilibrium properties of aluminum including its elastic constants, vacancy formation and migration energies, and surface energies. With accurate interatomic potentials, Nano-TO can be applied to design nanostructured materials using other base materials (e.g., copper, nickel, gold).”

3) In the supporting information, the nanoplate with a free surface $\{1\ 1\ 0\}$ is selected for comparison. How about other faces?

We thank the reviewer for this question. To answer the question, we have conducted additional simulations to consider another face and added discussions to the Supplementary Information:

“In addition to those two low-index surfaces, $\{1\ 0\ 0\}$ and $\{1\ 1\ 0\}$, there are oblique surfaces which are also perpendicular to the z -direction (i.e., $[0\ 0\ 1]$). Those oblique surfaces are more complex and often stepped surfaces. To consider oblique surfaces, an atomistic model of a nanoplate with a free surface $\{2\ 1\ 0\}$ on each side is created and shown in **Fig. S6a** (right). Similarly, periodic boundary conditions are imposed along the in-plane directions (i.e., $[\bar{1}\ 2\ 0]$ and $[0\ 0\ 1]$). Vacuum regions are created for both sides of the nanoplate in the $[2\ 1\ 0]$ direction to represent two free surfaces. In this specific model, the thickness is 2.72 nm. Other models with varying thicknesses are also created. The thickness of those models is in a range of 0.91 to 18.11 nm.”

“The modulus ratios in the z -direction for the nanoplates with varying thicknesses are shown in **Fig. S6b**. The modulus ratio is defined as the Young’s modulus of a nanoplate divided by that of the bulk material. The surface $\{1\ 0\ 0\}$ is stiffer than the surfaces $\{1\ 1\ 0\}$ and $\{2\ 1\ 0\}$ when the thickness is small. This result might explain why the surfaces in the Nano-TO designs are mostly $\{1\ 0\ 0\}$. As we only consider one type of oblique surface (i.e., $\{2\ 1\ 0\}$), future studies are required to confirm whether the surface $\{1\ 0\ 0\}$ is indeed the stiffest surface among all the possible surfaces (e.g., $\{3\ 1\ 0\}$, $\{3\ 2\ 0\}$) perpendicular to the z -

direction. It can be seen in the figure that the surfaces $\{1\ 0\ 0\}$, $\{1\ 1\ 0\}$, and $\{2\ 1\ 0\}$ are all softer than the bulk material due to the lower atomic coordination.”

4) The authors present the case studies of using Nano-TO in maximizing the bulk modulus and elastic constant. If the desired performance of nanostructured materials requires two or more improved mechanical properties, how to make use of this method and make balance in the design?

We thank the reviewer for this question. We have added discussions and a reference (ref. 35) about how Nano-TO could be applied to design nanostructured materials with multiple desired properties to the Discussions section:

“We show the applications of Nano-TO on the design of nanostructured materials for maximizing a desired property, in which two objectives, the bulk modulus and elastic constant of C_{33} (see the Supplementary Information), are considered, respectively. Various multi-objective optimization methods (ref. 35) can be implemented in Nano-TO to design nanostructured materials with multiple desired properties. The most common approach is the weighted sum method. To apply this method, the sensitivity values of each atom for different objectives are calculated individually and the weighted sensitivity value is calculated by choosing proper weights (user’s preference) for different objectives. Consequently, optimized designs with the best tradeoff between competing objectives can be generated by Nano-TO.”

5) As the authors mentioned that “the generality of the approach that it can be used to design nanomaterials and nanomachines with optimal performance”, it would be helpful to compare this method with prior approaches such as mentioned ML models, and describe how their method bases on the TO can be transferred to other nanomachines.

We thank the reviewer for this comment. To the best of our knowledge, there is no prior computational approach that can design nanostructured materials atom-by-atom as what Nano-TO can do. Thus, we cannot compare Nano-TO with other approaches in this study. However, as the reviewer mentioned, we believe that AI and ML techniques have the potential to accelerate the design process of Nano-TO. We are currently working on this research direction and have added discussions about it to the Discussion section:

“A typical material with a volume of a few cubic centimeters consists of around 10^{23} atoms. Even if the volume is reduced to a few cubic micrometers, the material still consists of around 10^{11} atoms. To achieve atomic-level precision, each atom is a design variable. Therefore, the computational cost to design materials on a scale of only a few micrometers is already beyond current computational capabilities. However, this computational bottleneck could be overcome in the future using AI and ML techniques. Future studies are required to develop suitable ML techniques to replace the computationally expensive atomistic modeling in the sensitivity analysis. We envision that this ML-based approach could potentially reduce the computational cost of Nano-TO by several orders-of-magnitude.”

Nanomachines can be considered as systems made up of different nanocomponents (or nanomaterials) and those nanocomponents would require different mechanical properties. We have added more information about how to apply Nano-TO for various mechanical properties and a reference (ref. 36) to the Discussion section:

“Although the examples presented here focus on elastic moduli, Nano-TO can be applied to design nanomaterials with other properties. It has been shown that the mechanical properties of metallic materials are highly related to their elastic moduli. For instance, an elastic anisotropy parameter can be used to identify alloys that display super elasticity, super strength, and high ductility, known as gum metals (ref.

36). If the correlations between the elastic moduli of a material and its other mechanical properties (e.g., failure strain, strength, toughness) can be discovered, the same design approach to tailor materials' elastic moduli can be applied to tailor materials' other mechanical properties.”

Reviewer 2

The work presents a method to optimize atomic structures. Though the algorithm is simple and straightforward, the concept is innovative. The following questions should be well answered before acceptance:

1) The process of optimization is shown in the flow chart in Fig. 1 but has not well been explained.

We appreciate the reviewer for this comment. We have added more details about the process of optimization to the Results section where Figure 1 is mentioned:

“Conceptually, the sensitivity analysis evaluates the contribution of each atom to the objective function (desired property) and this information is used to redistribute the atoms in the design domain. Afterwards, a sensitivity filtering technique is applied to modify the sensitivity value of each atom based on a weighted average of the sensitivity values of other atoms in a fixed neighborhood. The neighborhood region is defined by the filter radius. The purpose of applying the sensitivity filtering technique is to obtain the sensitivity values of “virtual” atoms (see Methods).”

“The number of “real” atoms to be converted to “virtual” atoms and that of “virtual” atoms to be converted to “real” atoms are controlled by the rejection and admission rates, respectively. As the initial structure consists of mostly “real” atoms, to reach the target volume fraction (relative density), the rejection rate has to be larger than admission rate in the first stage of the optimization process. The net rejection rate can be defined as the rejection rate minus admission rate. The smaller the net rejection rate, the more iterations are required to reach the target volume fraction. However, if the net rejection rate is too large, it can cause the optimization to converge to a low-quality design or make the optimization process unstable. After the target volume fraction is reached, the rejection and admission rates are set to be equal in the second stage of the optimization process, until the optimization is converged. Lastly, the atoms denoted by “real” will be kept and the atoms denoted by “virtual” will be removed from the design domain. Consequently, a nanostructured material with an optimized atom distribution can be generated.”

Additionally, we have added discussions of how the optimization parameters are selected and their effects in the optimization process to the Results section where the optimization setup is described:

“In the first stage, to ensure the stability of the optimization process, the rejection and admission rates are set to be 2 and 1, respectively. Consequently, two “real” atoms with the lowest sensitivity values are converted to “virtual” atoms while a “virtual” atom with the highest sensitivity value is converted to a “real” atom in each iteration, until the target volume fraction is reached. In the second stage, the rejection and admission rates are both set to be 1, until the optimization is converged. The filter radius is chosen to be slightly smaller than the lattice constant of the base material (i.e., aluminum), which is approximately 4 Å. Consequently, the 12 nearest neighbors of each atom are considered when the sensitivity filtering is implemented. Note that using a larger filter radius will cause the optimized designs to lose topological details (undesirable in this case) and increase the computational cost as more neighboring atoms have to be considered in the optimization process.”

2) A robust optimization algorithm should not rely on the initial structures. But the authors claim that 16 structures were produced in 16 attempts with random initialization. Then how the optimization guarantee the minimum of objective functional, no matter locally or globally.

We thank the reviewer for this great question. We have added discussions and a reference (ref. 29) about nonconvex optimization problems to the Discussion section.

“Most TO problems at the continuum-scale, except for some simple cases, are nonconvex optimization problems, which contain many local minima (ref. 29). We find that this is also the case in the Nano-TO examples presented in this study since the optimized designs are generally dependent on the initial structure. We show that Nano-TO is capable of identifying the optimal designs by using several different initial structures. However, this dependence on the initial structure could become a computational bottleneck when applying Nano-TO to large-scale materials design problems as the computational cost is increased. Therefore, further improvements of the Nano-TO approach are essential to make it less sensitive to the initial structure and prevent generating low-quality designs.”

Additionally, we have provided an explanation of why optimized designs generated by Nano-TO would depend on the initial structures to the Supplementary Methods section:

“To ensure that the optimized designs are of high performance, Nano-TO is performed multiple times starting with different initial structures. The only difference between those initial structures is the location of the initially assigned “virtual” atom. Although those initial structures are physically identical due to the symmetry of the crystal structure and periodic boundary conditions, they would lead to different optimized designs. The reason is that multiple atoms would have the same sensitivity value during the optimization process. The ranking of those atoms with the same sensitivity value will retain their initial order (the atom index) in the sensitivity analysis. Consequently, different initial structures would lead to different optimization paths and converge to different optimized designs.”

3) Is the surface effect considered in atomistic modeling? If yes, how to take it into consideration?

We thank the reviewer for this question. Yes, the surface effect is considered in our atomistic simulations. We have added more details and a reference (ref. 30) to the Methods section about how we take the surface effect into consideration in our simulations:

“Pair potentials (e.g., Lennard–Jones) do not include the environmental dependence of bonding. Therefore, the strength of individual bonds in the bulk is the same as that on (or near) the surface, which is physically not true. This local environmental dependence is especially important for simulations of surfaces and can be considered in many-body potentials such as the EAM potentials (ref. 30).”

“Since the embedding energy term considers the local background electron density of atoms, the EAM potentials can describe the variation of bond strength with coordination. Therefore, the EAM potentials are applicable for modeling material systems with surfaces or other crystalline defects as those investigated in this study.”

4) How to impose periodic boundary condition?

We thank the reviewer for this question. The periodic boundary conditions are imposed by using the boundary command in LAMMPS. We have added a conceptual description of periodic boundary

conditions to the Supplementary Methods section and included a link of the LAMMPS website for readers to find more details about this function:

“Periodic boundary conditions are imposed by using the style p boundary command in LAMMPS (<http://lammmps.sandia.gov>). Consequently, each atom in the unit cell not only interacts with the other atoms in the same unit cell but also with their mirror images in the adjacent unit cells.”

5) How to determine the embedding energy and potential energy in Eq. (4).

We thank the reviewer for this question. The embedding energy and potential energy in Eq. (4) are functions of the distance r_{ij} between atom i and j . Thus, with a give set of the distance r_{ij} , the embedding energy and potential energy can be calculated with Eq. (4). We have added more information about how those potential functions are developed to the Methods section:

“The potential functions of the embedding energy F_α , atomic density ρ_β , and pair-wise potential energy $\phi_{\alpha\beta}$ in the EAM potential are fitted to both experimental data and *ab initio* calculations to accurately reproduce basic equilibrium properties of aluminum including its elastic constants, vacancy formation and migration energies, and surface energies.”

6) The statement “an energy minimization using the conjugate gradient (CG) algorithm is performed to equilibrate the material system” is very unclear. Please give more details to show how to calculate the elastic properties? In homogenization theory when calculating the effective properties like C11, only one test strain is imposed. But the work needs a positive strain and a negative test strain, why? Will the magnitude of test strain affect the value of C11?

We thank the reviewer for this comment. We have added more details about the elastic properties’ calculations to the Supplementary Methods section:

“To calculate the elastic properties of a material system (e.g., Nano-TO design, gyroid structure), the system is first relaxed to reach the equilibrium state (stress-free). An energy minimization using the conjugate gradient (CG) algorithm is performed to equilibrate the system. During the energy minimization, the simulation box is allowed to adjust the size and the atoms are allowed to move to reduce the total energy of the system. After the equilibrium state is reached, a small negative strain (-10^{-3}) is applied along the x -direction ($\epsilon_{xx} = -10^{-3}$) and all other strains are set to be zero. In this deformed state, the corresponding stress tensor is computed after the system is relaxed by another energy minimization. As the strains of the system are fixed, the simulation box is not allowed to adjust the size, however, the atoms are still allowed to move to reduce the total energy of the system during the energy minimization. The elastic constant of C_{11} can be calculated from the stress tensor as the ratio between the stress and the applied strain along the x -direction. The elastic constant of C_{11} can also be calculated from another stress tensor by applying a small positive strain (10^{-3}) along the x -direction ($\epsilon_{xx} = 10^{-3}$).”

Additionally, in the Code availability section, we have provided a link for the script used for calculating the elastic properties. To address the concern about elastic properties calculated in compression and tension, we have added discussions about it to the Supplementary Methods section:

“The elastic constants are formally defined in terms of the second derivative of the free energy with respect to strain evaluated at zero strain. By definition, these are independent of the sign of the strain. However, in our computational approach, we apply finite strains to evaluate the elastic constants. The implication is that higher order terms in the Taylor series defining the free energy at this strain also contribute to the

observed stress. This can lead to small differences depending on the sign of the imposed strain. For instance, when a strain of 10^{-3} is applied to calculate the elastic constants of aluminum, the elastic constant of C_{11} is calculated as 113.94 GPa in compression and 113.57 GPa in tension. The values are very close, and the difference is only 0.3%. To average these values, we fit a line to the stress–strain curve, and the slope of this line is used to determine the elastic constant of interest.”

To address the concern about different values of the elastic constants could be obtained when using different magnitudes of the applied strain, we have added discussions about it to the Supplementary Methods section:

“Furthermore, we find that the elastic constants thus determined are insensitive to the magnitude of the applied strain when the strain is small enough (smaller than 10^{-2}). For instance, the elastic constant of C_{11} (mean value) is calculated as 113.76 GPa for aluminum when the applied strain is 10^{-3} . This value becomes 113.79 GPa when the applied strain is decreased to 10^{-4} or 113.39 GPa when the applied strain is increased to 10^{-2} .”

Reviewer 3

The paper presents a so-called nano-topology optimization method for atom-by-atom controlled material design. While the paper is well written and well described the method, the reviewer didn't find it interesting. In particular, I couldn't find any novelty and broader impacts of the paper. It seems the authors are trying to re-invent the wheel that is already out there—by using the same old-fashioned optimization approach, just at a different scale. More importantly, people in the design and optimization community have already tackled these kinds of problems in many different ways and at different length scales. A small survey on the designing (and even manufacturing) of architected and cellular material with desired properties (linear/nonlinear) would reveal those well-established approaches. Therefore, I don't see any novelty and broader impact by re-trying a well-established old-fashioned topology optimization approach at another length scale. To me, this is just a matter of scaling, and you could get the same results (even better) at the continuum level. You could also develop any of these objectives (even complex cases) at the continuum level and obtain any desired properties, as people have done this before. In conclusion, the reviewer does not recommend the manuscript for publication in Nat. Comm.

We appreciate the reviewer for the comments. To emphasize the differences between Nano-TO and the conventional TO approaches using FEM and to avoid the misunderstanding that the same designs can be obtained at the continuum level, we have added more discussions about the advantages of Nano-TO over the conventional TO approaches to the Discussion section:

“We demonstrate that Nano-TO can utilize the surface effect in the design of nanostructured materials. By optimizing the surface topology at the nanoscale, the HS upper bound for the bulk modulus can be exceeded. Note that the surface effect comes from the variation of bond strength with coordination, which can only be captured in atomistic simulations. Therefore, the conventional TO approaches using FEM will not generate the Nano-TO designs shown in this study.”